# Hyperspectral Image Spectral–Spatial Classification Method Based on Deep Adaptive Feature Fusion

**Caihong Mu** [1], **Yijin Liu** [1] **and Yi Liu** [2,*]

1  Key Laboratory of Intelligent Perception and Image Understanding of Ministry of Education, Collaborative Innovation Center of Quantum Information of Shaanxi Province, International Research Center for Intelligent Perception and Computation, Joint International Research Laboratory of Intelligent Perception and Computation, School of Artificial Intelligence, Xidian University, Xi'an 710071, China; caihongm@mail.xidian.edu.cn (C.M.); yijinliu@stu.xidian.edu.cn (Y.L.)
2  School of Electronic Engineering, Xidian University, Xi'an 710071, China
*  Correspondence: yiliu@xidian.edu.cn

**Abstract:** Convolutional neural networks (CNNs) have been widely used in hyperspectral image (HSI) classification. Many algorithms focus on the deep extraction of a single kind of feature to improve classification. There have been few studies on the deep extraction of two or more kinds of fusion features and the combination of spatial and spectral features for classification. The authors of this paper propose an HSI spectral–spatial classification method based on deep adaptive feature fusion (SSDF). This method first implements the deep adaptive fusion of two hyperspectral features, and then it performs spectral–spatial classification on the fused features. In SSDF, a U-shaped deep network model with the principal component features as the model input and the edge features as the model label is designed to adaptively fuse two kinds of different features. One comprises the edge features of the HSIs extracted by the guided filter, and the other comprises the principal component features obtained by dimensionality reduction of HSIs using principal component analysis. The fused new features are input into a multi-scale and multi-level feature extraction model for further extraction of deep features, which are then combined with the spectral features extracted by the long short-term memory (LSTM) model for classification. The experimental results on three datasets demonstrated that the performance of the proposed SSDF was superior to several state-of-the-art methods. Additionally, SSDF was found to be able to perform best as the number of training samples decreased sharply, and it could also obtain a high classification accuracy for categories with few samples.

**Keywords:** hyperspectral image classification; adaptive feature fusion; multi-feature fusion; multi-scale and multi-level feature extraction model





## 1. Introduction

A hyperspectral sensor is a spectrometer that can simultaneously image a specific area on consecutive tens or hundreds of bands to obtain a hyperspectral image (HSI). Compared with multispectral images, HSIs have a wide range of bands and higher spectral resolution. Because hyperspectral imaging involves different bands, HSIs can obtain rich spectral information [1], which is conducive to resource exploration [2] and environmental monitoring [3]. However, due to its high data dimensions, there is a problem of dimensional disaster in HSI processing. In fact, in the classification of hyperspectral data, many bands are redundant and have little positive effect on the classification result, so they seriously affect the processing results and efficiency. Therefore, feature selection and feature extraction came into being. For example, principal component analysis (PCA) [4,5] and independent component analysis (ICA) [6,7] are typical methods that transform high-dimensional data into low-dimensional data. In traditional HSI classification methods, support vector machine (SVM) [8,9], random forest [10], and other methods have been considered as efficient algorithms. Moreover, a problem is that different spectra presented by HSIs may belong to

the same category and similar spectra may belong to different categories, so it is difficult to obtain a high accuracy in classification by only considering spectral information. In recent years, the question of how to make full use of spatial features has become attractive in the field of HSI classification.

Kang et al. [11] used the first principal component or the first three principal components of an HSI as a gray or color guide image to perform an edge-preserving filtering on the probability map obtained by the classifier, and then they selected the largest probability pixel to achieve classification. Additionally, adding texture features can be used to increase the classification accuracy of an HSI [12]. In recent years, deep networks have gained widespread attention. A stacked autoencoder (SAE) [13,14], as one of the typical deep learning models, can extract and classify features by encoding and decoding the input vectors. Deep belief networks (DBNs) [15] and convolutional neural networks (CNNs) [16–18] have been proposed for spectral–spatial HSI classification. Furthermore, to obtain deep-level features, Zhao et al. [19] used dimensionality reduction methods and 2DCNN models to extract spectral and spatial features. Using the neighborhood block as the input of the network, a 3DCNN [20,21] was used to direct extracted spectral and spatial features from an original HSI to make full use of its spectral–spatial features and improve the classification results. Mou et al. [22] proposed the idea to use the time-series networks such as the RNN (recurrent neural network), LSTM (long short-term memory), and GRU (gated recurrent unit) for HSI classification, but the method only extracted hyperspectral spectral features, thus leading to limited classification accuracy. Xu et al. [23] proposed a multi-scale CNN model. This model first performed PCA on HSIs to extract three principal components as the input of the network, which combined the characteristics of each pooling layer with the spectral characteristics to classify HSIs. In this method, only three principal components were extracted from hyperspectral data as input features, and most HSI information was lost, so it was not good enough to achieve excellent classification results. Zhong et al. [24] designed a spectral spatial residual network (SSRN) for HSI classification, where the input data was a three-dimensional cube and the network used spectral and spatial residual blocks to learn discriminative features from the rich spectral and spatial features in the original HSI. However, this method only used residual alternating learning to obtain fusion features, and the feature fusion was not sufficient, so the extraction of spatial features was not good enough. Mu et al. [25] proposed a multi-scale and multi-level spectral–spatial feature fusion network (MSSN), where neighborhood blocks of different scales were used as the input of the network. The spectral features extracted by the 3D convolutional neural network and the spatial features extracted by the 2D convolutional neural network were combined in the form of 3D–2D alternating residual blocks and a self-mapping method. Song et al. [26] designed a deep feature fusion network (DFFN) for HSI classification by introducing residual learning and simultaneously adding the outputs of different levels of networks to further improve the classification results. The fusion method, however, was only an addition operation of the output features at different levels, the feature fusion of which was too simple and resulted in an insufficient fusion. Guo et al. [27] proposed an efficient deep feature extraction and HSI classification method based on multi-scale spatial features and cross-domain convolutional neural network (MSCNN) that could make full use of the multi-scale spatial features obtained by the guided filter. The cross-domain convolutional neural network was used to reorder the multi-scale spatial features, which were then input into a simple convolutional neural network model for classification. This method only performed a kind of recombination operation on the edge features, and it did not introduce other features. The network model only extracted features simply and did not make full use of the features of the HSI.

To solve the above problems and to adaptively fuse the two different features in a deep network, we propose a spectral–spatial HSI classification method based on deep adaptive feature fusion (SSDF). In this paper, a U-shaped network structure was used to enhance the fusion of deep features. The edge features and the principal component features of the HSIs were fused adaptively to obtain new features. The new features were input into a

multi-scale and multi-level feature extraction (MMFE) model, and the output features were then combined with the spectral features for classification.

The contributions of this work are as follows:

(1) The authors of this paper propose a U-shaped deep network that can adaptively fuse two different features consisting of convolutional layers, pooling layers, and deconvolution layers. The U-shaped network model was constructed to make sure that the two different features and the new feature after fusion have the same size. The labels of the U-shaped network are not the true labels of the image that are used in most literature; instead, they are the edge feature maps obtained by the guided filtering. The inputs of the U-shaped network are hyperspectral principal component feature maps. The U-shaped network is trained to learn the correlation and complementarity of two different features, adaptively fusing two different features and generating new feature maps. The new feature maps alleviate the problem of the low classification accuracy caused by using single kind of features.

(2) The authors of this paper designed an MMFE model that extracts the feature map of each pooling layer for convolution operation and finally inputs the convolved features to the global average pooling layer to extract the main information. The extraction of multi-level and multi-scale features can deeply extract the edge and abstract features of the image, which is beneficial to the final classification. The proposed deep adaptive feature fusion and spectral–spatial classification network uses advanced and different kinds of features as the input of the classification network, which can realize the multi-scale and multi-level fusion of multiple features, thus resulting in higher classification accuracy.

## 2. Materials and Methods

Here, we introduce SSDF. Sections 2.1–2.3 introduce the methods for feature extraction and fusion, and Sections 2.4–2.6 introduce the methods for the classification of the extracted features. Assume there is a hyperspectral dataset $X = \{x_1, x_2, \ldots, x_N\} \in \mathbb{R}^{1 \times 1 \times b}$, where $N$ is the number of labeled pixels and $b$ is the number of spectral bands. $Y = \{y_1, y_2, \ldots, y_N\} \in \mathbb{R}^{1 \times 1 \times L}$ represents the corresponding one-hot label vector set, where $L$ is the category of objects. We partition all data available into three sets—the training, validation, and test sets, which are denoted by $Z^1$, $Z^2$, and $Z^3$, respectively. Their corresponding one-hot label vector sets are $Y^1$, $Y^2$, and $Y^3$. First, the SSDF network uses $Z^1$ and $Y^1$ to update the network parameters. Then, $Z^2$ and $Y^2$ are used to monitor the temporary model generated by the network. Finally, $Z^3$ and $Y^3$ are used to evaluate the performance of the optimal training model.

### 2.1. Guided Filter for Edge Feature Extraction

Guided filtering [28] is an edge preservation filter with excellent performance that can make the output image retain the characteristics of the filtered image and better load the edge information of the guided image. In fact, the guided filtering method makes use of a local linear relationship between the output image of the guided filtering and the guided image. Assuming that the guided image is $I$ and the input image is $s$, the filtered output image $c$ is obtained by a local linear model as follows:

$$c_i = a_k I_i + d_k \ \forall i \in \omega_k, c_i \in c \tag{1}$$

where $\omega_k$ is a square window with pixel $k$ at the center and its length and width is $(2r + 1)$. $a_k$ and $d_k$ are the coefficients to be estimated of the linear model.

Then, to estimate the $a_k$ and $d_k$ parameters of the linear model, a cost function is established according to the difference between the input image $s$ and the output image $c$:

$$E(a_k, d_k) = \sum_{i \in \omega_k} \left( (a_k I_i + d_k - s_i)^2 + \varepsilon a_k^2 \right) \tag{2}$$

where $\varepsilon$ is a regularization parameter that avoids making $a_k$ too large. Finally, the ridge regression technique [29] is used for parameter estimation. By minimizing the cost function (Equation (2)), the coefficients $a_k$ and $d_k$ can be solved as follows:

$$a_k = \frac{1}{|\omega|} \frac{\sum\limits_{i \in \omega} I_i s_i - \mu_k \bar{s}_k}{\sigma_k^2 + \varepsilon} \tag{3}$$

$$d_k = s_k - a_k \mu_k \tag{4}$$

where $\mu_k$ and $\sigma_k^2$ are the mean and the variance of the guided image $I$ in the window, respectively; $|\omega|$ is the total number of pixels in the window; and $\bar{s}_k$ is the mean of the input image $s$ in the window. The guided filtered output image can be calculated after the $a_k$ and $d_k$ coefficients are obtained.

It can be seen from Equation (1) that the output image and the guided image have a linear relationship in the window, that is $\Delta c_i = a_k \Delta I_i$. Therefore, when the guided image $I$ contains edge information, the output image $c$ retains the edge information at the corresponding position. Therefore, the output image $c$ is a feature map with the edge features of the HSI.

The authors of this paper used guided filtering technology to extract the edge features of the image, as shown in Figure 1.

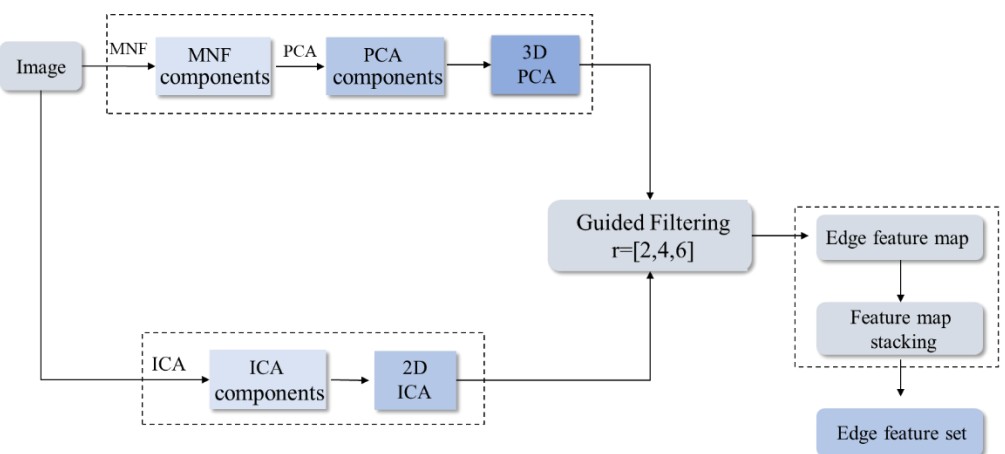

**Figure 1.** Flow chart of guided filtering to extract edge features. PCA: principal component analysis; ICA: independent component analysis; MNF: minimum noise fraction rotation.

In Figure 1, it can be seen that the minimum noise fraction rotation (MNF) [30] is used to denoise the input image first, and then PCA is used to extract the first few principal components of the denoised image as the input image $PC_1$-$PCe$ for guided filtering. The guided image is the first independent component feature map $IC_1$ of the HSI extracted by ICA. Taking $PC_1$-$PCe$ as input images and using $IC_1$ and three different windows [2,4,6] to perform guided filtering operations to obtain $3e$ filtering feature vectors at various scales, we can stack all vectors to form a multi-scale guided filtering feature set, namely the edge feature image set.

### 2.2. Principal Component Feature Extraction

Because the high-dimensional characteristics of HSIs bring problems such as computational complexity and information redundancy, it was required to use PCA to reduce the dimensionality of the spectral information of HSIs and to extract the first $e$ principal

component features. The spectral matrix $X_s$ of the HSI is obtained according to the spectral information of the samples as follows:

$$X_s = \begin{pmatrix} x_{11}, x_{12}, \cdots, x_{1p} \\ x_{21}, x_{22}, \cdots, x_{2p} \\ \vdots \\ x_{n1}, x_{n2}, \cdots, x_{np} \end{pmatrix} \tag{5}$$

where $n$ denotes the number of all pixels in the HSI, $p$ denotes the length of the spectral information of the samples, $X_s$ represents the spectral matrix of the HSI with $n$ samples, and each row of $X_s$ represents a spectrum sample with length $p$. We calculate the average value $\overline{x}_i$ of the $i$-th dimensional spectral information of the sample by the following formula:

$$\overline{x}_i = \frac{1}{n} \sum_{k=1}^{n} x_{ki} \tag{6}$$

Further calculations yield the covariance matrix $S$ of the spectral matrix $X_s$:

$$S = \begin{pmatrix} S_{11}, \cdots, S_{1j}, \cdots, S_{1n} \\ \vdots \ \vdots \ \vdots \\ S_{i1}, \cdots, S_{ij}, \cdots, S_{in} \\ \vdots \ \vdots \ \vdots \\ S_{n1}, \cdots, S_{nj}, \cdots, S_{nn} \end{pmatrix} \tag{7}$$

The component at the $i$-th row and the $j$-th column of the covariance matrix $S$ is:

$$S_{ij} = \frac{1}{n-1} \sum_{k=1}^{n} (x_{ki} - \overline{x}_i) \bullet \left( x_{kj} - \overline{x}_j \right) \tag{8}$$

where $x_{kj}$ represents the $j$-th dimensional spectral value of the $k$-th sample, $\overline{x}_j$ represents the average value of the $j$-th dimensional spectral values of all the samples, and $1 < k \leq n$.

Then, the covariance matrix $S$ is diagonalized, and the feature vectors are orthogonally normalized. The normalized eigenvectors are arranged according to the size of the corresponding eigenvalues, from large to small, to obtain a feature matrix $X_z$. Then the spectral feature matrix $X_a = X_z * X_s$, where the first $c$ columns of $X_a$ are the first $c$ principal component features of the HSI.

Thus far, we used guided filtering to obtain the features that contain the main edge information of the HSI, and we adopted PCA to reduce the dimensions to obtain the features that contain the principal components of an HSI.

### 2.3. Adaptive Feature Fusion

During the linear transformation of guided filtering, due to the difference in the radius of the sliding window, a part of the image information will be lost and the image information will not be fully utilized. In contrast, the principal component features of an HSI are the first few principal components of the image obtained by the PCA dimensionality reduction of the whole image, which can make up for the problem of information loss caused by the different sliding window radii in guided filtering. Therefore, to make more comprehensive use of HSI information, the authors of this paper adaptively fused these two different HSI features so that the HSI information could be fully utilized. As a deep autonomous learning model, deep learning can make a network adaptively learn the correlation and difference between model inputs and model labels by training on a network, thereby generating fusion features that contain both edge features and principal component features. Thus, the authors of this paper designed a U-shaped deep fusion network model with the principal component features as the model input and the edge

features as the model label. The final output of the model comprises the fusion features that contain edge features and principal component features. The U-shaped network structure is shown in Figure 2.

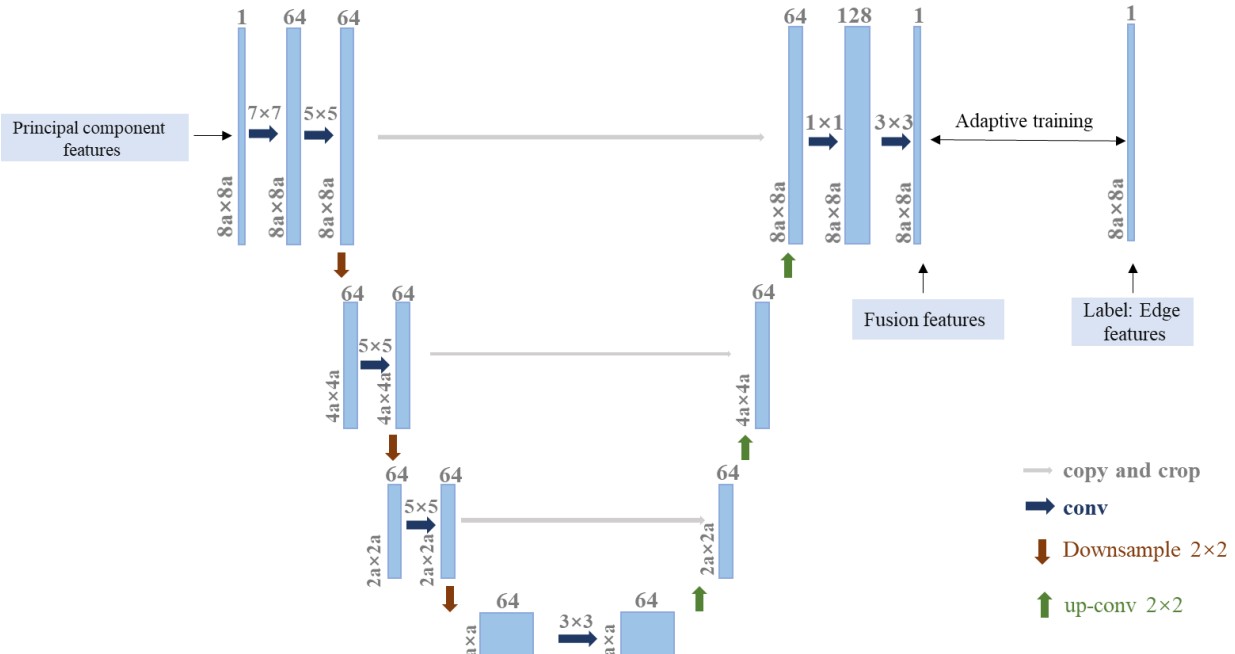

**Figure 2.** U-shaped deep adaptive fusion model.

In Figure 2, a, 2a, 4a, and 8a represent the size of the feature map; $1 \times 1$, $3 \times 3$, $5 \times 5$, and $7 \times 7$ represent the size of the convolution kernel; and 1, 64, and 128 represent the dimensions of the image. The fusion model consists of three parts: a convolution layer, a pooling layer (downsampling layer), and a deconvolution layer. This U-shaped structure, using the redefined input data and label data, enhances the fusion of the two features. As a feature extractor, the convolution layer can convert the input image into multi-scale features, which makes the features more abstract. However, the purpose of the designed network is to fuse features with the same size as the input features. Therefore, a deconvolution layer is designed after the convolutional layer, which can generate dense and enlarged feature maps.

Assume that the input data of the U-shaped model is $x$, and the output of the $i$-th convolutional layer is represented as:

$$F_i(x) = \theta_i(\mu_i) \tag{9}$$

$$\mu_i = F_{i-1}(x) * \mathbf{w_i} + b_i \tag{10}$$

where $\theta_i(\cdot)$ is the activation function of the $i$-th layer and $\mathbf{w_i}$ and $b_i$ represent the filters and bias vectors of the $i$-th layer, respectively. According to the description of the proposed U-shaped architecture, the estimation of the network parameters $\Theta = \{\mathbf{w_i}, b_i | i \in (1, 2, \ldots, M)\}$ can be obtained by minimizing the loss between the fusion features and the label features, where $M$ is the number of layers of the model. The loss function is expressed by mean square error as follows:

$$Loss(x, \Theta) = \frac{1}{H} \sum_{h=1}^{H} ||F(x_h, \Theta) - x_h^{GF}||_2^2 \tag{11}$$

where $F(x_h, \Theta)$ represents the output of the network, $x_h^{GF}$ represents the label feature map, and $H$ represents the number of pixels on each feature map. The authors of this paper made use of multiple feature maps for feature fusion, so after each training gets a fused feature, the network parameters are initialized and the next feature map is retrained until all feature maps have been trained. All the obtained fusion feature maps are stacked in the spectral dimension to obtain the adaptive fusion features with the same dimensions and sizes as the original two features. These features include the edge and principal component features of the HSI, and the fusion features obtained by the adaptive method are more beneficial to the HSI classification.

### 2.4. Multi-Scale and Multi-Level Feature Extraction

The method shown in Figure 2 merely fuses the two kinds of features adaptively. To achieve better classification results, we needed to design a deep-level feature extraction and classification model. In this paper, an MMFE model was designed for classification. With the increasing of the number of convolutional layers, the spatial size of the feature map decreases sharply, leading to some information loss of the image. In a traditional CNN architecture, the fully connected layer is usually directly connected to the output of the last convolutional layer. In this case, the network pays more attention to the deep features and ignores the shallow features. The authors of this paper propose combining shallow convolution features with deep ones in classification. A diagram of the MMFE is given in Figure 3.

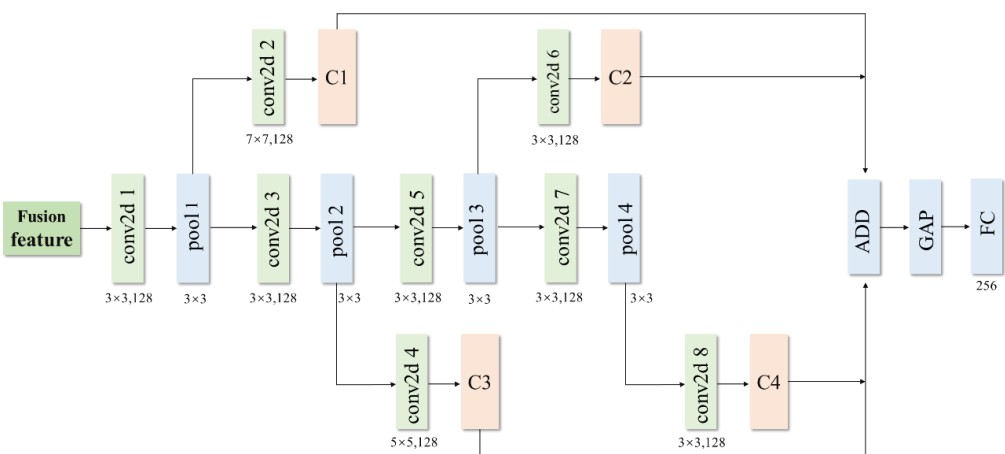

**Figure 3.** Multi-scale and multi-level feature extraction (MMFE) model. GAP: global average pooling; FC: fully connected layer.

To make full use of the features of different levels, the proposed network adds a 2D convolution layer (conv2d) after each pooling layer. The first purpose of this is to extract multi-level features by adding convolution layers at different levels. The second is that the size of the feature map can be changed by using a convolution layer so that the feature maps of different levels have the same size after passing through the convolution layer. Let $C^i = f(w^i x^i + b^i)$ denote the $i$-th feature map obtained by introducing the convolutional layer after the pooling layer, where $f$ is the activation function, $x^i$ is the feature map after the pooling layer, and $w^i$ and $b^i$ are the corresponding weight matrices and bias terms, respectively. The multi-level feature map output by the convolution is input to the ADD layer to perform addition $C^5 = \sum_{i=1}^{4} C^i$, and the combined $C^5$ is then input to the global average pooling (GAP) to stretch it into a one-dimensional tensor, where the GAP can extract the main information of the feature map and can reduce parameters at the same time. The output of GAP finally passes through the fully connected layer (FC). This model extracts the deep features of the new features after fusion, and the obtained features also have multiple levels.

### 2.5. Spectral Feature Extraction

The authors of this paper used the LSTM model to extract the spectral features of the original HSI. The core module in the LSTM model is the storage unit, as shown in Figure 4.

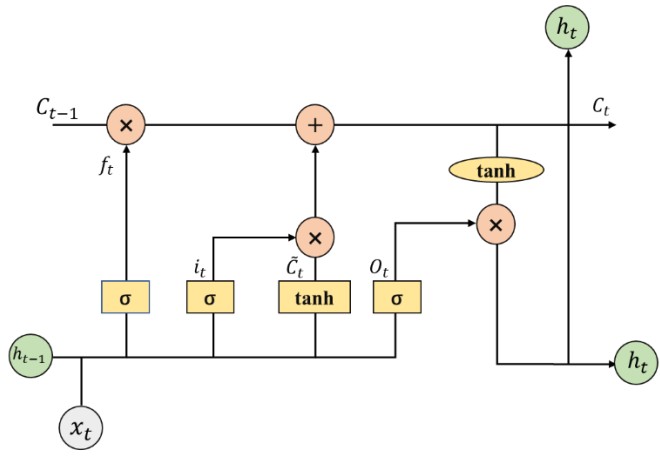

**Figure 4.** LSTM (long short-term memory) storage unit.

The storage unit consists of four elements, i.e., the input gate ($i_t = \sigma(R_i x_t + G_i h_{t-1} + b_i)$), the forget gate ($f_t = \sigma(R_f x_t + G_f h_{t-1} + b_f)$), the output gate ($o_t = \sigma(R_o x_t + G_o h_{t-1} + b_o)$), and the cell state ($C_t = i_t \otimes \tanh(R_c x_t + G_c h_{t-1} + b_c) + f_t \otimes C_{t-1}$). The input gate determines how much new information is added to the cell state ($C_t = i_t \otimes \tanh(R_c x_t + G_c h_{t-1} + b_c) + f_t \otimes C_{t-1}$). The output of the output gate is based on the cell state, but it is also a filtered version. The forget gate determines what information is discarded from the cell sate. The output of LSTM is $h_t = o_t \otimes \tanh(C_t)$. In the above formulas, $R_i, R_f, R_o, R_c, G_i, G_f, G_o, G_c$ is the weight matrix and $b_i, b_f, b_o, b_c$ is the bias vector; $\tan h$ is the hyperbolic tangent; $\sigma(x) = 1/(1 + \exp(-x))$ is the sigmoid function; and $\otimes$ is the dot product. $x$ is each band of the HSI input into the LSTM model, and $h$ is the corresponding one-dimensional resultant vector. All the bands of pixels on the HSI are sequentially input into the LSTM model, and then a one-dimensional vector is output. Thus, the one-dimensional vector is a feature vector with the spectral characteristics of the HSI.

### 2.6. SSDF Model

Sections 2.1–2.4 introduced the adaptive feature fusion of edge features and principal component features, as well as the MMFE model for classification. These methods focus on processing the spatial context of pixels without considering the correlation between the pixels and different bands. Though deep feature extraction and fusion are performed on HSIs, most of the band information is ignored during feature preprocessing, and spectral features are not fully utilized. Therefore, we further introduced the LSTM model (Section 2.5) to extract the spectral band information of the HSI, which is combined with the spatial features obtained by the MMFE model to perform spectral–spatial classification and form a complete SSDF method. As shown in Figure 5, SSDF designs a deep adaptive fusion model to fully fuse the principal component features extracted by PCA and the edge features extracted by guided filtering, and the fused features are further extracted by MMFE and then combined with the spectral features extracted by LSTM for the final classification, which improves the classification accuracy. The Pavia University hyperspectral dataset [31] was input into the network of the proposed SSDF as an example, which is shown in Figure 5.

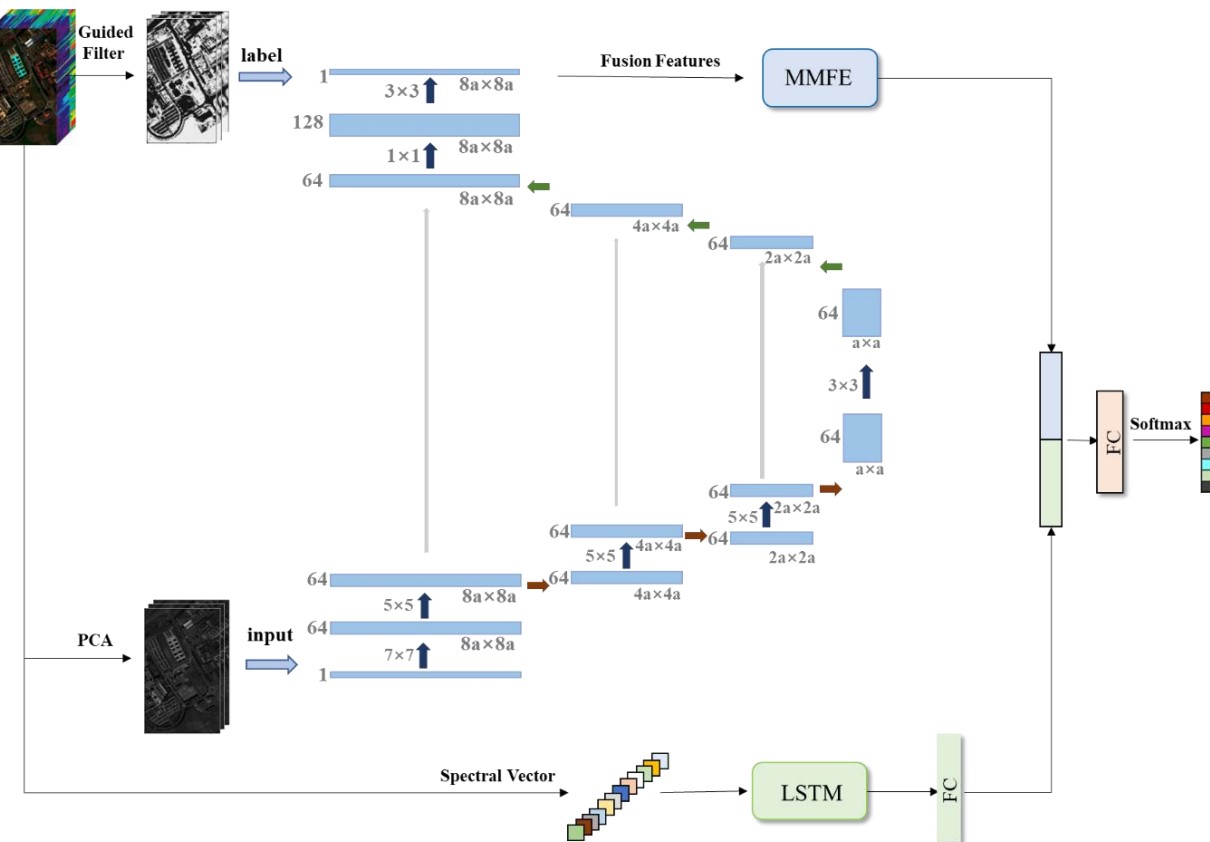

**Figure 5.** Spectral–spatial classification method based on deep adaptive feature fusion (SSDF) model.

The network in the upper half of Figure 5 consists of two parts: one is the feature fusion network and the other is the MMFE model. The two networks are independently trained without affecting each other. In the feature fusion network, the first *c* principal components of the HSI are used as the input of the fusion network, and the hyperspectral edge feature map obtained by the guided filtering is used as the labels of the fusion network. During network training, only one principal component feature map is input at a time, corresponding to the label that is also an edge feature map. The feature map output by the last layer of the network after training is a fusion feature map that combines two hyperspectral features. A total of *c* feature maps need to be fused, so each time a feature map is re-input, the network parameters are initialized to ensure training consistency. After all the *c* feature maps are trained, the obtained *c* feature maps are stacked to get the final fusion feature. Then, the obtained fused features are input into the MMFE network to further extract multi-scale and multi-level features. The MMFE network outputs a one-dimensional spatial feature vector $\hat{y}_a$ by using convolution layers and pooling operations alternately, followed by a fully connected layer.

The input of the lower half of Figure 5 is the original HSI. For this example, all the bands of each pixel of the HSI were input into the LSTM model to obtain the one-dimensional feature vector of the pixel. This feature vector was then input into a fully connected layer to further extract integrated features to obtain a spectral feature vector $\hat{y}_b$.

In the SSDF model, the spatial feature vector $\hat{y}_a$, spectral feature vector $\hat{y}_b$, and the classifier training are integrated into a unified network. To complete the unified spectral–spatial classification using the feature stacking method, the feature vector $\hat{y}_a$ obtained in the MMFE is connected to the feature vector $\hat{y}_b$ obtained in the LSTM to form a new feature

vector $\hat{y}$, which then passes through a fully connected layer and a SoftMax layer. The loss function of SSDF is defined as Equation (12).

$$L = -\frac{1}{m}\sum_{i=1}^{m}[y_i\log(\hat{y}_i) + (1 - y_i)\log(1 - \hat{y}_i)] \tag{12}$$

where $y_i$ represents the label feature map, $\hat{y}_i$ represents the corresponding predicted label of the *i*-th training sample, and *m* represents the size of the training set. As the classification network is trained, all parameters are simultaneously optimized by a small batch random gradient descent algorithm. Finally, the SoftMax layer generates the prediction vector set $\hat{y} = \{\hat{y}_1, \hat{y}_2, \ldots, \hat{y}_N\}$.

## 3. Experiment Setup

### 3.1. Experimental Datasets

We used three real hyperspectral datasets [31,32] to test the performance of SSDF, including the Indian Pines, Pavia University, and Salinas scene datasets.

(1) Indian Pines dataset: This was captured by the Airborne Visible/Infrared Imaging Spectrometer (AVIRIS) from the remote sensing test area in the northwest area of the Indian state. This dataset contained 220 bands, and 20 noise bands were removed before the experiments. Each band was of size 145 × 145 and had a wavelength ranging from 0.4 to 2.5 μm. It contained 16 ground-truth classes and a total of 10,249 samples with a spatial resolution of 20 m per pixel. Figure 6a,b shows its false-color image and the corresponding ground-truth map.

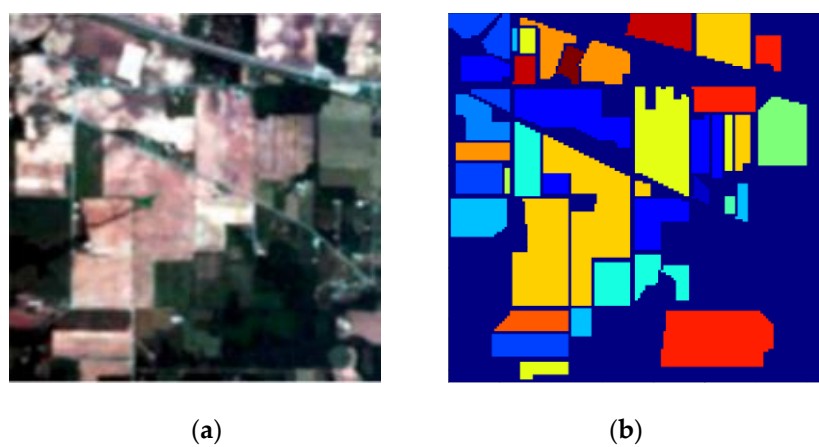

(**a**)                    (**b**)

**Figure 6.** Indian Pines dataset. (**a**) The false-color image. (**b**) The ground-truth map.

(2) Pavia University dataset: This was captured by the Reflective Optics System Imaging Spectrometer (ROSIS) from Pavia University in northeastern Italy. This dataset contained 115 bands, and 12 noise bands were removed before the experiments. Each band was of size 610 × 340 and had a wavelength ranging from 0.43 to 0.86 μm. It contained nine ground-truth classes and a total of 42,776 samples with a spatial resolution of 1.3 m per pixel. Figure 7a,b shows its false-color image and the corresponding ground-truth map.

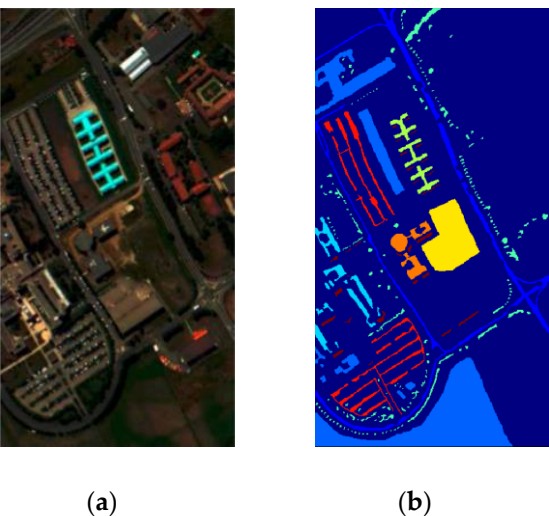

<center>(<b>a</b>)　　　　　　　　　　　　(<b>b</b>)</center>

**Figure 7.** Pavia University dataset. (**a**) The false-color image. (**b**) The ground-truth map.

(3) Salinas scene dataset: This was collected by AVIRIS sensor over Salinas Valley, California. This dataset contained 224 bands, and 20 water absorption bands were removed before the experiments. Each band was of size 512 × 217. It contained 16 ground-truth classes and a total of 54,129 samples with a spatial resolution of 3.7 m per pixel. Figure 8a,b shows its false-color image and the corresponding ground-truth map. Table 1 introduces the meaning of each category in three datasets and the number of samples they contained.

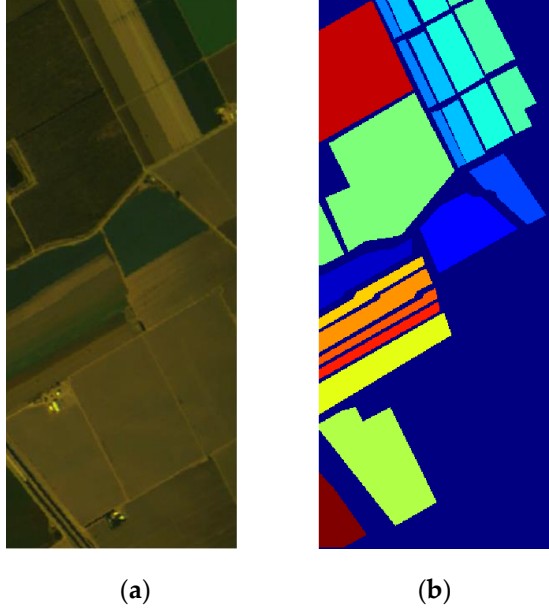

<center>(<b>a</b>)　　　　　　　　　　　　(<b>b</b>)</center>

**Figure 8.** Salinas scene dataset. (**a**) The false-color image. (**b**) The ground-truth map.

**Table 1.** Information of each category in three datasets.

| | Indian Pines | | | | Pavia University | | | | Salinas Scene | | |
|---|---|---|---|---|---|---|---|---|---|---|---|
| No. | Color | Class | Samples | Color | Class | Samples | Color | Class | Samples | | |
| 1 | | Alfalfa | 46 | | Asphalt | 6631 | | Brocoli_G_W_1 | 2009 | | |
| 2 | | Corn-N | 1428 | | Meadows | 18,649 | | Brocoli_G_W_2 | 3726 | | |
| 3 | | Corn-M | 830 | | Gravel | 2099 | | Fallow | 1976 | | |
| 4 | | Corn | 237 | | Trees | 3064 | | Fallow_R_P | 1394 | | |
| 5 | | Grass-M | 483 | | P-M-sheets | 1345 | | Fallow_smooth | 2678 | | |
| 6 | | Grass-T | 730 | | Bare Soil | 5029 | | Stubble | 3959 | | |
| 7 | | Grass-P-M | 28 | | Bitumen | 1330 | | Celery | 3579 | | |
| 8 | | Hay-W | 478 | | S-B-Bricks | 3682 | | Grapes_untrained | 11,271 | | |
| 9 | | Oats | 20 | | Shadows | 947 | | Soil_V_D | 6203 | | |
| 10 | | Soybean-N | 972 | | | | | Corn_S_G_W | 3278 | | |
| 11 | | Soybean-M | 2455 | | | | | Lettuce_R_4wk | 1068 | | |
| 12 | | Soybean-C | 593 | | | | | Lettuce_R_5wk | 1927 | | |
| 13 | | Wheat | 205 | | | | | Lettuce_R_6wk | 916 | | |
| 14 | | Woods | 1265 | | | | | Lettuce_R_7wk | 1070 | | |
| 15 | | Build-G-T-D | 386 | | | | | Vinyard_untrained | 7268 | | |
| 16 | | Stone-S-T | 93 | | | | | Vinyard_V_T | 1807 | | |

### 3.2. Experimental Setup

We divided all samples of each dataset into training, validation, and test sets. Then, we used the training set to update the parameters, used the validation set to monitor the generation of the network temporary model, and kept the model with the highest validation rate. Finally, we used the test set to test the classification performance of the reserved model. For Indian Pines dataset, 10%, 10%, and 80% samples were randomly selected from each class as the training, validation, and test sets, respectively. For the Pavia University and Salinas scene datasets, 5%, 5%, and 90% samples were randomly selected from each class as the training, validation, and test sets, respectively.

Overall accuracy (OA), average accuracy (AA), and the kappa coefficient were adopted as the evaluation indicators [33] to evaluate the classification performance of each method. In our experiments, after appropriate experimental adjustments, the training epoch of the fusion network was set to 2000 times, and the learning rate was set to 0.003. The training epoch of the MMFE and LSTM network was set to 400 times, and the batch size was set to 128. The pooling layer used the maximum pooling operation. For this paper, the activation function at the output layer used the SoftMax activation function, and the activation functions in other locations all used the Rectified Linear Units (ReLU) activation function. For the Indian Pines, Pavia University, and Salinas scene datasets, the numbers of guided filtering input images were 10, 5, and 5, respectively, and the guided filtering radii of the three datasets were 2, 4, and 6, respectively. Thus for the three datasets, we got fused feature dimensions of 30, 15, and 15, respectively. To avoid bias estimates, we ran the experiments five times and provided the final results by calculating the average of five values. All experiments were performed on the NVIDIA 1080Ti graphics card using Python.

*3.3. Influence of Settings*

3.3.1. The Effectiveness of Fusion Features

To verify the validity of fusion features, we compared the experimental results of fusion features with those of single kind of features on the three datasets. As shown in Table 2, the principal component features (PFs), edge features (EFs), and fusion features (FFs) proposed in this paper were input into the MMFE, and they were then combined with the spectral characteristics obtained by the LSTM model for classification. The effects of different features were compared in terms of OA, AA, and kappa coefficient.

**Table 2.** The results of different features on different datasets. OA: overall accuracy; AA: average accuracy; PFs: principal component features; EFs: edge features; FFs: fusion features.

| Indian Pines (10%) | PFs | EFs | FFs |
|---|---|---|---|
| OA | 98.53% | 98.50% | 99.37% |
| AA | 95.32% | 97.19% | 99.08% |
| Kappa | 0.9833 | 0.9829 | 0.9929 |
| **Pavia University (5%)** | **PFs** | **EFs** | **FFs** |
| OA | 98.87% | 99.39% | 99.68% |
| AA | 98.68% | 99.41% | 99.49% |
| Kappa | 0.9850 | 0.9927 | 0.9958 |
| **Salinas Scene (5%)** | **PFs** | **EFs** | **FFs** |
| OA | 98.51% | 99.13% | 99.81% |
| AA | 99.28% | 99.41% | 99.79% |
| Kappa | 0.9834 | 0.9903 | 0.9979 |

It can be seen from Table 2 that the results of fusion feature were better than those of the single kind of features on the three evaluation indicators. For example, in the Indian Pines dataset, if only the hyperspectral principal component features or edge features were used for classification, the AA was only 95.32% or 97.19%, respectively. After fusing two kinds of features, the AA of the fusion features reached 99.08%, and the accuracy was improved by about 4% or 2%, respectively, which showed that the fused features performed well in the classification tasks. For the Pavia University and Salinas scene datasets, the results of FFs were also better than those of PFs and EFs. The results shown in Table 2 demonstrate that the idea of feature fusion is effective, with focus on the complementarity between different features. Thus, the multi-feature fusion had a richer correlation between spectral information and spatial information, thus improving the classification performance of the network.

3.3.2. The Effectiveness of Introducing Spectral Features by LSTM

To verify the necessity and effectiveness of introducing spectral features by LSTM, three classification methods were compared: the method that directly inputs HSIs into LSTM (LSTM), the method that only uses MMFE classification after the U-shaped network that generates fusion features (U-shaped and MMFE), and the SSDF method proposed in this paper. The results are given in Table 3. It can be seen in Table 3 that the SSDF method had the best overall classification accuracy on the three datasets.

**Table 3.** The overall accuracy (%) of different methods.

| Methods | Indian Pines | Pavia University | Salinas Scene |
|---|---|---|---|
| LSTM | 81.57 | 90.99 | 92.64 |
| U-shaped and MMFE | 98.96 | 99.04 | 99.28 |
| SSDF | 99.37 | 99.68 | 99.81 |

It can be seen from Table 3 that if only LSTM was used for classification, the classification results were not satisfactory. Because the extraction of single spectral features only shallowly uses the information of the HSI, the use of the spatial relationship between pixels and the reasonable judgment of edge pixels was insufficient. The second method was found to significantly improve the classification results by introducing principal component features, edge features, and a multi-scale and multi-level classification structure. However, this method mainly focuses on processing the spatial context of pixels, ignoring the correlation between the pixels and different bands. In contrast, SSDF was found to achieve better results by combing the features of two methods. Experimental results showed that the further introduction of spectral features by LSTM was necessary, and it further improved classification performance.

## 4. Classification Results

We compared the proposed method with two classic methods, the SVM [8,9] and PCA [4,5], and five state-of-the-art methods, LSTM [22], 3DCNN [21], SSRN [24], DFFN [26], and MSCNN [27].

The SVM, as a classic machine learning method for classification, was used as a baseline for comparison. The PCA method used here referred to using the first 20 principal components and SVM with Radial Basis Function (RBF-SVM) for classification. LSTM [22] used long-term and short-term memory models to extract and utilize spectral features in image bands. 3DCNN [21] used a three-dimensional convolution cube to extract spectral and spatial features from the original HSIs. SSRN [24] fused the spectral features obtained by the 3D convolution kernel and the spatial features obtained by the 2D convolution kernel in a tandem manner, allowing the model to obtain the spectral and spatial features. DFFN [26] combined features extracted from residual networks in different levels for classification. MSCNN [27] proposed the use of cross-domain convolutional neural networks for feature extraction and classification. For the sake of fairness, we adjusted the parameters to make these comparison methods achieve their best performances, and we trained these models in the exact same experimental environment.

### 4.1. Results on Indian Pines

Table 4 shows the classification results of eight methods on the Indian Pines dataset. Figure 9 shows the false-color image, the ground truth, and the classification maps of all methods on the Indian Pines dataset.

As can be seen from Table 4, the OA results of SSDF were 18.19% and 22.98% higher than those of SVM and PCA, respectively. The OA values of SSDF were also 17.8%, 8.99%, 1.15%, 0.31%, and 0.86% higher than the LSTM, 3DCNN, SSRN, DFFN, and MSCNN state-of-the-art methods, respectively. It can be observed from Figure 9 that the classification maps of SVM, PCA, LSTM, and 3DCNN had the serious problem of "salt and pepper", whereas the classification maps of DFFN and our SSDF were most similar to the ground truth. Additionally, the SSDF method still performed well when there were few samples in some categories. As shown in Table 4, the classification accuracy of SSDF on the first category (Alfalfa) reached 100%, and it reached 100% on the seventh category (Grass-P-M), which exceeded most other classification algorithms. At the same time, SSDF could also obtain satisfactory results in the 9th and 16th categories.

**Table 4.** The classification results of all methods on the Indian pine dataset with OA, AA and Kappa data given in the form of mean ± standard deviation. Legend: SVM, support vector machine; PCA, principal component analysis and SVM with Radial Basis Function (RBF-SVM); LSTM, long short-term memory network; 3DCNN, three-dimensional convolutional neural network; SSRN, spectral–spatial residual network; DFFN, deep feature fusion network; MSCNN, multi-scale spatial features and cross-domain convolutional network; and SSDF, spectral–spatial classification method based on deep adaptive feature fusion.

| Class | SVM | PCA | LSTM | 3DCNN | SSRN | DFFN | MSCNN | SSDF |
|---|---|---|---|---|---|---|---|---|
| 1 | 83.33 | 61.90 | 56.10 | 59.52 | 97.37 | 100 | 80.95 | **100** |
| 2 | 72.78 | 63.22 | 75.18 | 91.60 | 99.35 | 98.88 | 97.82 | **99.73** |
| 3 | 65.19 | 58.50 | 66.80 | 87.01 | 97.62 | 99.41 | **100** | 99.24 |
| 4 | 63.08 | 45.33 | 61.97 | 85.98 | 79.98 | 100 | **100** | 98.85 |
| 5 | 90.57 | 89.66 | 86.41 | 88.51 | 98.98 | 98.43 | 97.24 | **99.48** |
| 6 | 95.59 | 96.96 | 95.28 | 98.93 | 98.66 | **99.99** | 99.85 | 99.48 |
| 7 | 69.23 | 50.00 | 68.00 | 84.62 | 90.48 | 100 | 92.31 | **100** |
| 8 | 93.51 | 98.61 | 97.91 | 100 | 100 | 100 | 100 | **100** |
| 9 | 72.22 | 22.22 | 66.66 | 94.44 | 94.44 | 80 | **100** | 94.12 |
| 10 | 71.09 | 67.09 | 82.15 | 84 | 97.64 | **99.51** | 98.51 | 98.58 |
| 11 | 86.11 | 79.64 | 82.71 | 91.04 | 98.66 | 98.18 | 98.19 | **99.70** |
| 12 | 72.28 | 64.98 | 72.23 | 76.03 | **98.32** | 98.15 | 94.57 | 97.65 |
| 13 | 96.76 | 95.68 | 100 | 99.46 | 100 | 100 | **100** | 99.38 |
| 14 | 97.89 | 96.22 | 96.75 | 97.54 | 98.33 | 100 | 99.82 | **100** |
| 15 | 47.71 | 49.14 | 44.96 | 77.87 | 100 | 99.98 | 99.43 | **100** |
| 16 | 80.95 | 84.52 | 92.78 | 97.62 | **98.65** | 95.83 | 98.81 | 97.22 |
| OA (%) | 81.18 ± 1.54 | 76.39 ± 0.89 | 81.57± 1.05 | 90.38 ± 0.89 | 98.22 ± 0.25 | 99.06 ± 0.16 | 98.51 ± 0.31 | **99.37 ± 0.12** |
| AA (%) | 78.64 ± 1.49 | 70.23 ± 1.35 | 80.62± 1.52 | 88.39 ± 1.46 | 98.08 ± 0.59 | 98.24 ± 0.50 | 97.34 ± 0.57 | **99.08 ± 0.17** |
| Kappa × 100 | 78.36 ± 1.02 | 72.94 ± 1.12 | 78.92 ± 0.48 | 89.03 ± 0.96 | 97.97 ± 0.85 | 98.93 ± 0.23 | 98.29 ± 0.36 | **99.29 ± 0.11** |

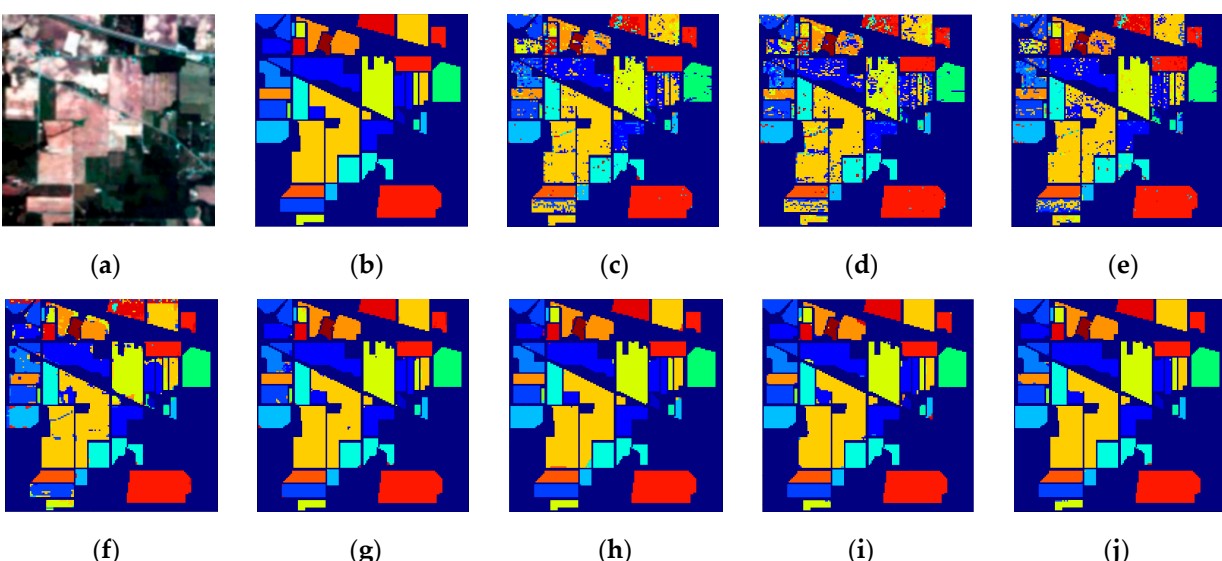

**Figure 9.** Classification maps for the Indian Pines dataset. (**a**) False-color image. (**b**) Ground truth. (**c**) SVM. (**d**) PCA. (**e**) LSTM. (**f**) 3DCNN. (**g**) SSRN. (**h**) DFFN. (**i**) MSCNN. (**j**) SSDF.

### 4.2. Results on Pavia University

Table 5 provides the classification results of eight methods on the Pavia University dataset. Figure 10 shows the false-color image, the ground truth, and the classification maps of all methods on the Pavia University dataset.

As can be seen from Table 5, the OA values of SSDF were 6.3% and 6.51% higher than those of the traditional SVM and PCA methods, respectively. The OA values of SSDF were also 7.04%, 3.42%, 0.17%, 0.41%, and 0.62% higher than the LSTM, 3DCNN, SSRN, DFFN, and MSCNN state-of-the-art methods, respectively. At the same time, it was seen that SSDF achieved better classification results than other methods in most categories. It can be observed from Figure 10 that the classification maps of SSRN, DFFN, MSCNN, and SSDF were very close to the ground truth. In contrast, other methods had the serious problem of "salt and pepper."

**Table 5.** The classification results of all methods on the Pavia University dataset with OA, AA, and Kappa data given in the form of mean ± standard deviation.

| Class | SVM | PCA | LSTM | 3DCNN | SSRN | DFFN | MSCNN | SSDF |
|---|---|---|---|---|---|---|---|---|
| 1 | 92.87 | 94.33 | 91.67 | 95.84 | 99.94 | 99.78 | 98.60 | **99.97** |
| 2 | 98.09 | 98.49 | 97.00 | 98.79 | 100 | **100** | 99.99 | 99.96 |
| 3 | 74.03 | 78.65 | 78.89 | 87.42 | 94.21 | 98.79 | 94.74 | **98.94** |
| 4 | 94.68 | 92.51 | 92.04 | 96.39 | **98.26** | 96.55 | 98.42 | 98.04 |
| 5 | 99.37 | 98.90 | 98.67 | 99.14 | 99.51 | 99.01 | **100** | 99.7 |
| 6 | 85.72 | 81.56 | 89.06 | 91.27 | 100 | 100 | 100 | **100** |
| 7 | 83.07 | 82.28 | 79.50 | 93.35 | 99.58 | 99.49 | 98.49 | **100** |
| 8 | 90.74 | 88.94 | 86.27 | 95.77 | 99.43 | 98.46 | 97.37 | **99.61** |
| 9 | 99.78 | 99.89 | 99.65 | 97 | **100** | 90.49 | 96.78 | 98.72 |
| OA (%) | 93.38 ± 0.68 | 93.17 ± 0.46 | 92.64 ± 0.64 | 96.26 ± 0.18 | 99.51 ± 0.09 | 99.27 ± 0.16 | 99.06 ± 0.23 | **99.68 ± 0.08** |
| AA (%) | 90.93 ± 0.54 | 90.62± 0.67 | 90.99± 0.83 | 94.99 ± 0.80 | 99.49 ± 0.17 | 98.63 ± 0.52 | 98.27 ± 0.32 | **99.49 ± 0.06** |
| Kappa × 100 | 91.18 ± 0.82 | 90.87 ± 0.23 | 90.23± 0.56 | 95.03 ± 0.50 | 97.97 ± 0.12 | 99.03 ± 0.19 | 98.76 ± 0.31 | **99.58 ± 0.11** |

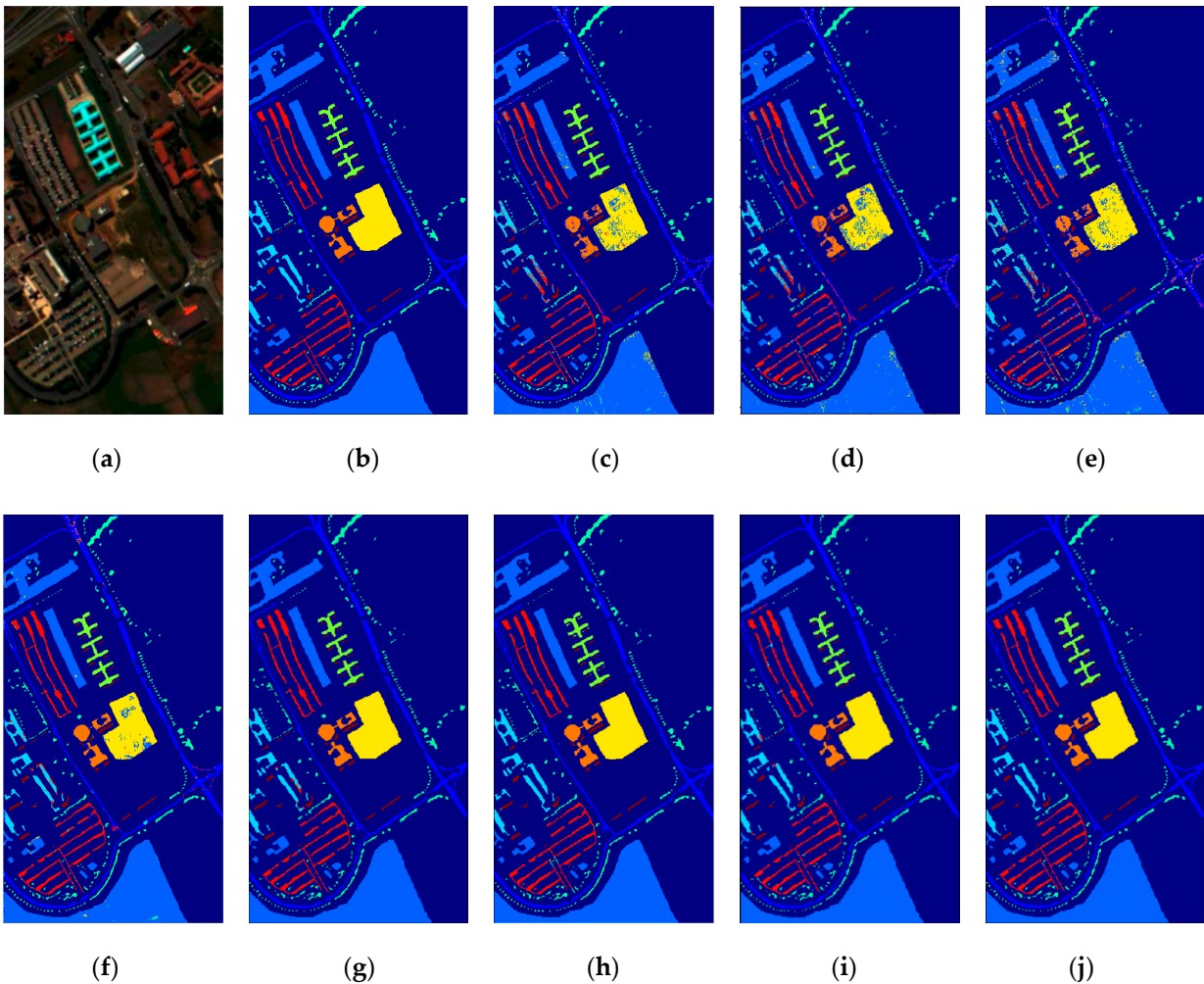

**Figure 10.** Classification maps for the Pavia University dataset. (**a**) False-color image. (**b**) Ground truth. (**c**) SVM. (**d**) PCA. (**e**) LSTM. (**f**) 3DCNN. (**g**) SSRN. (**h**) DFFN. (**i**) MSCNN. (**j**) SSDF.

### 4.3. Results on Salinas Scene

Table 6 gives the classification results of eight methods on the Salinas scene dataset. Figure 11 shows the false-color image, the ground truth, and the classification maps of all methods.

**Table 6.** The classification results of all methods on the Salinas scene dataset with OA, AA, and Kappa data given in the form of mean ± standard deviation.

| Class | SVM | PCA | LSTM | 3DCNN | SSRN | DFFN | MSCNN | SSDF |
|---|---|---|---|---|---|---|---|---|
| 1 | 96.8 | 98.32 | 99.67 | 95.29 | 100 | 100 | 100 | **100** |
| 2 | 97.01 | 99.97 | 99.94 | 99.80 | 100 | 100 | 100 | **100** |
| 3 | 99.31 | 97.23 | 99.09 | 99.57 | 100 | 100 | 98.62 | **100** |
| 4 | 98.04 | 99.55 | 99.51 | 97.58 | 99.84 | 99.52 | 97.51 | **99.84** |
| 5 | 96.58 | 97.99 | 98.80 | 99.84 | 99.29 | **99.96** | 99.49 | 99.30 |
| 6 | 95.93 | 99.65 | 99.92 | 99.04 | 100 | 100 | 100 | **100** |
| 7 | 97.03 | 99.38 | 99.51 | 97.94 | 100 | **100** | 99.94 | 99.97 |
| 8 | 82.96 | 90.94 | 91.34 | 93.68 | 98.39 | 98.73 | 99.26 | **99.57** |
| 9 | 98.79 | 99.89 | 99.69 | 98.61 | 100 | 100 | 100 | **100** |
| 10 | 87.12 | 93.80 | 93.97 | 94.09 | 100 | 99.39 | 99.84 | 99.83 |
| 11 | 91.23 | 91.23 | 95.84 | 94.98 | 99.16 | 97.51 | 99.21 | **100** |
| 12 | 98.96 | 99.95 | 99.19 | 99.62 | 100 | 99.54 | 100 | **100** |
| 13 | 93.69 | 97.24 | 99.02 | 96.9 | 100 | **100** | 99.31 | 99.88 |
| 14 | 85.55 | 94.39 | 93.78 | 94.99 | 98.76 | **99.37** | 98.03 | 98.96 |
| 15 | 69.04 | 58.07 | 62.85 | 87.31 | 95.72 | 98.33 | 95.83 | **99.82** |
| 16 | 89.34 | 98.66 | 99.20 | 95.34 | 99.69 | 99.87 | 99.18 | **100** |
| OA (%) | 89.35 ± 0.43 | 91.38±0.23 | 92.38±0.73 | 95.56 ± 0.79 | 99.01 ± 0.23 | 99.38 ± 0.08 | 99.04 ± 0.28 | **99.81 ± 0.05** |
| AA (%) | 92.34 ± 0.36 | 94.77±0.83 | 95.96±0.64 | 96.54 ± 0.64 | 99.57 ± 0.15 | 99.51 ± 0.19 | 99.14 ± 0.12 | **99.79 ± 0.08** |
| Kappa × 100 | 88.09 ± 0.71 | 90.37±0.33 | 91.49±0.85 | 95.06 ± 0.51 | 98.89 ± 0.31 | 99.31 ± 0.11 | 98.93 ± 0.18 | **99.79 ± 0.06** |

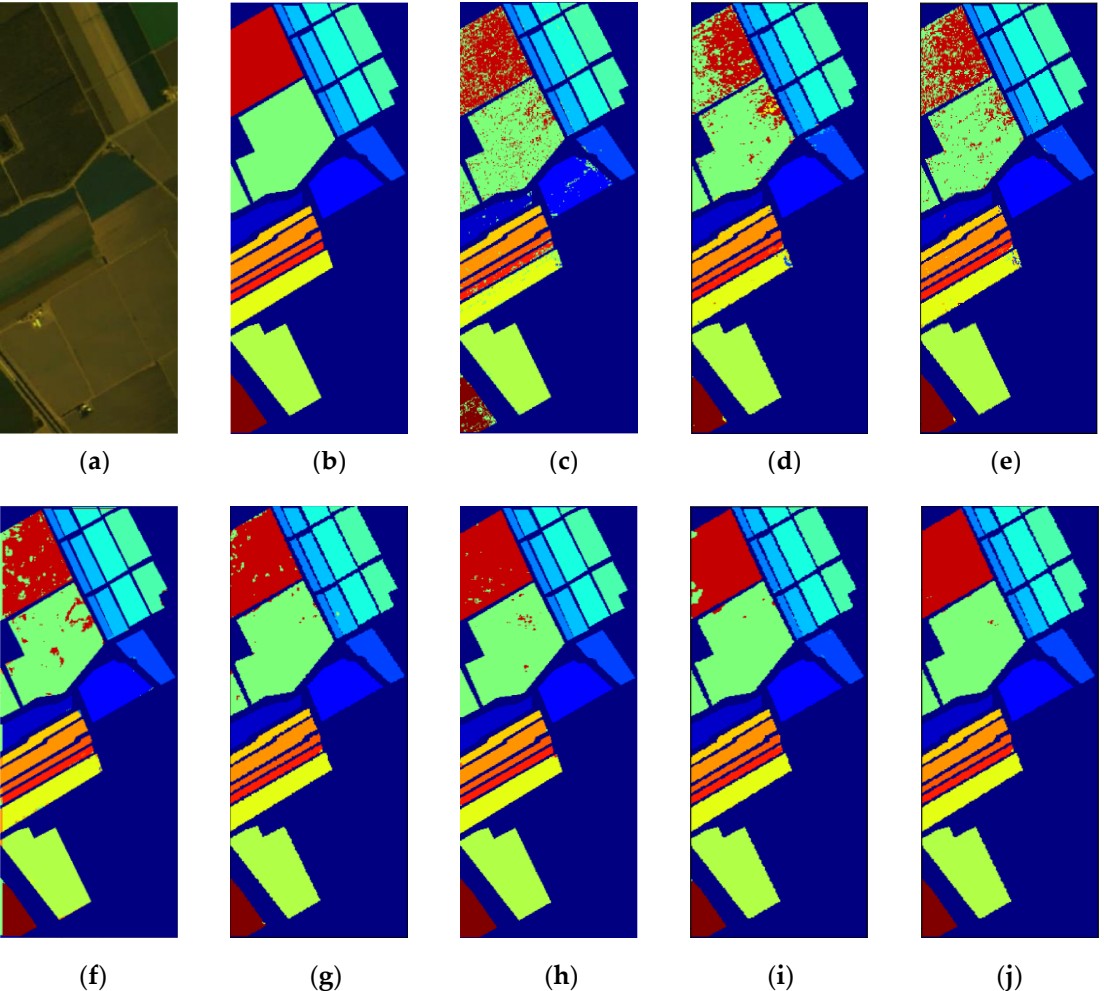

**Figure 11.** Classification maps for the Salinas scene dataset. (**a**) False-color image. (**b**) Ground truth. (**c**) SVM. (**d**) PCA. (**e**) LSTM. (**f**) 3DCNN. (**g**) SSRN. (**h**) DFFN. (**i**) MSCNN. (**j**) SSDF.

It can be seen from Table 6 that the OA values of SSDF were 10.46% and 8.43% higher than those of the traditional SVM and PCA methods, respectively. The OA values of SSDF were also 7.43%, 4.25%, 0.8%, 0.43%, and 0.77% higher than LSTM, 3DCNN, SSRN, DFFN, and MSCNN, respectively. Specifically, it can be seen that for the 8th (Grapes_untrained) and 15th classes (Vinyard_untrained), which were difficult for classification, SSDF achieved the best results. At the same time, according to the variance values of the evaluation indexes in Table 6, it can be seen that the performance of SSDF was more stable. It can be observed from Figure 11 that the classification map of SSDF was much more similar to the ground truth than those of other methods. Especially, the red area in the upper left corner of each image in Figure 11 distinctly demonstrates the superiority of SSDF.

## 5. Discussion

To test the generalization ability and robustness of the SSDF, we randomly selected 5%, 10%, 15%, and 20% labeled samples from the Indian Pines dataset and 3%, 4%, 5%, and 6% labeled samples from the Pavia University and Salinas scene datasets as the training data. The curves in Figure 12 show the overall accuracies of the eight methods versus different percentages of training samples.

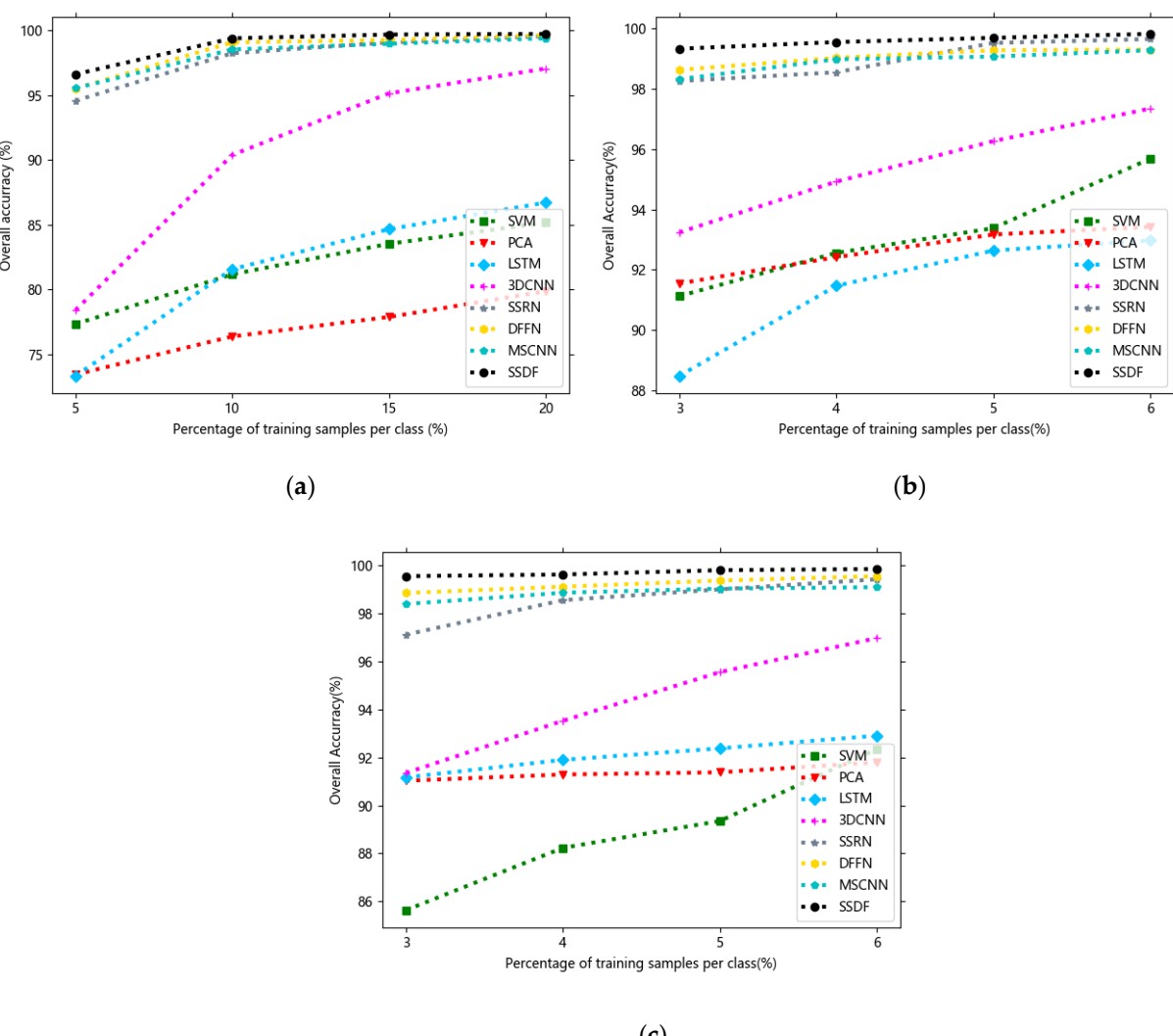

**Figure 12.** The curves of overall accuracies versus different percentages of training samples obtained by different methods on three datasets: (**a**) Indian Pines dataset, (**b**) Pavia University dataset, and (**c**) Salinas scene dataset.

It can be seen from Figure 12 that when there were fewer training data, SSDF could still achieve a much higher classification accuracy than SVM, PCA, LSTM, and 3DCNN, and it was also superior to other the state-of-the-art SSRN, DFFN and MSCNN methods.

As can be seen from all the above experimental results, the proposed SSDF achieved the best classification performance in most categories, and it also obtained the best classification results on the three evaluation indicators of OA, AA, and Kappa. The reasons for this performance improvement are as follows:

(1) The principal component features and edge features were used as the input and label of the U-shaped network, respectively, so that the network adaptively generated new fusion features that could adaptively learn the correlation and complementarity of two different features through network training and provide more sufficient information for classification.

(2) The MMFE model combined low-level features with high-level features, making the model perform better.

Compared with the simple spectral–spatial combination of SSRN, the proposed SSDF introduced the idea of merging multiple features, thus fully merging the rich feature correlation and feature dissimilarity between two different features. At the same time, compared with the MSCNN network, SSDF not only used the U-shaped network to adaptively generate advanced features but also introduced the idea of a multi-scale and

multi-level classification network, and this structure could more deeply extract the original information of HSIs and further improve the classification effect.

## 6. Conclusions

The authors of this paper have proposed a hyperspectral image spectral–spatial classification based on deep adaptive feature fusion (SSDF). Compared with other existing network models, the U-shaped structure in SSDF is composed of special inputs and labels, i.e., the principal component features and edge features are used as the input and label of the U-shaped network, respectively. Corresponding training of inputs and labels through deep networks can effectively extract and fuse two elementary features to generate advanced features. Moreover, compared with a network model with single feature input, SSDF was found to greatly retain the complementarity and rich correlation among the features by making full use of various features. Additionally, the proposed SSDF model contains a multi-scale and multi-level network for extracting deep features that, to some extent, fuses elementary features with advanced features, thus making our method more generalizable. The experimental results showed that the performance of SSDF on the three datasets was better than other existing state-of-the-art methods, and SSDF was always able to obtain good classification results under different training conditions, which further validated that the proposed SSDF has excellent generalization ability and robustness.

Though the idea of multi-feature fusion brings higher classification accuracy, it also increases the computational complexity of a model. In the future, we will try to simplify the proposed model. Furthermore, although the fusion of edge features and principal component features has shown its effectiveness of improving classification accuracy, we will investigate the possibility of working with other types of features for fusion, which may result in better performance if better combination of features is found.

**Author Contributions:** Conceptualization, Y.L. (Yijin Liu); formal analysis, Y.L. (Yijin Liu); funding acquisition, C.M. and Y.L. (Yi Liu); investigation, Y.L. (Yijin Liu); methodology, Y.L. (Yijin Liu); project administration, C.M.; software, Y.L. (Yijin Liu); supervision, C.M.; validation, C.M.; visualization, Y.L. (Yijin Liu); writing—original draft, Y.L. (Yijin Liu); writing—review and editing, C.M. and Y.L. (Yi Liu). All authors have read and agreed to the published version of the manuscript.

**Funding:** This research was funded by the National Natural Science Foundation of China under grants 62077038, 61672405, 61876141, U1701267, 61772399, 61773304, and 61802295.

**Data Availability Statement:** Publicly available datasets were analyzed in this study. This data can be found here: [http://www.ehu.eus/ccwintco/index.php?title=Hyperspectral_Remote_Sensing_Scenes] (accessed on 23 January 2021).

**Acknowledgments:** The authors would like to thank the editor and anonymous reviewers who handled our paper.

**Conflicts of Interest:** The authors declare no conflict of interest.

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
