# Peer review of "Hyperspectral Image Spectral–Spatial Classification Method Based on Deep Adaptive Feature Fusion"

_remotesensing, doi:10.3390/rs13040746_

Round 1

Reviewer 1 Report

The presentation of the paper is good, and the scientific methods used in the paper are sound. Only very minor text editing is required as follows:

  1. line 147 :  ' ε is a regularization ---'  --> 'where ε is a regularization ---'
  2. line 150 :  ' μk  and  ---'  --> 'where μk  and  ---'

Reviewer 2 Report

The overall research design is appropriate; materials and methods are clearly described. Minor spell check is required

Reviewer 3 Report

Please see PDF attached.

Reviewer 4 Report

The paper is well-organized and the topic is a hot area of research. Here are some of my concerns: 

  • Why the authors used PCA? The denoising autoencoder or other deep architectures can be used for data dimension reduction. Did the authors explore this possibility?
  • How did the Authors set the parameters of the networks? Filter size, number of layers, etc. Is this an optimized architecture? 
  • In Table 4, it has been shown that the accuracy in some classes is 100. But, in some others not. What metrics and criteria can distinguish the slight difference in accuracy? (in terms of visual assessments)
  • Figure 11 shows the class map of different methods. The last 3 methods are very competitive with each other. How the authors distinguish these three methods? 

The paper is written very well and the language is good. 

Author Response

This manuscript is a resubmission of an earlier submission. The following is a list of the peer review reports and author responses from that submission.